# Coping Processes of Patients with Ostomies in South Korea: A Focus Group Study [note 1]

**DOI:** 10.3390/healthcare9010021

**Published:** 2020-12-27

**Authors:** Heesook Son, Youngmi Kang

**Affiliations:** 1Red Cross College of Nursing, Chung-Ang University, Seoul 06974, Korea; hson@cau.ac.kr; 2East-West Nursing Research Institute, College of Nursing Science, Kyung Hee University, Seoul 02447, Korea

**Keywords:** coping, mental suffering, nursing, ostomy, physical suffering, psychological adaptation, psychological distress, psychological sexual dysfunction, social adjustment

## Abstract

Despite the adverse effects of negative coping after receiving a stoma, there is a lack of information on how patients cope with ostomies and on their families’ experiences throughout the coping process. We aimed to explore the coping experiences of individuals with ostomies throughout their illness, applying the Corbin and Strauss Chronic Illness Trajectory Framework, using exploratory qualitative methods involving focus group interviews. Purposive sampling was utilized to recruit 19 participants (split across three groups) through an ostomy association in South Korea. Two focus group interviews were separately conducted from March through to May 2017 until data saturation was achieved. Using qualitative content analysis, we analyzed the transcribed interviews and identified words and themes to interpret the results. The coping experiences of patients with ostomies were expressed through three interrelated themes: struggling and suffering, learning how to live with ostomy, and living with ostomy. We found gender differences in spousal support and a struggle among older patients regarding social coping. The themes identified suggest that gender should be considered when designing interventions to help patients cope with ostomies.

## 1. Introduction

An ostomy is a surgical opening on the abdomen created for the elimination of urine or feces due to various medical conditions [1]. The worldwide increase in the incidence of colorectal cancers and inflammatory bowel diseases has led to an increase in ostomy creation surgeries [2], making the quality of life of patients with an ostomy a global health issue. In South Korea, approximately 26,790 new cases of colorectal cancer are diagnosed per year [3]. In addition, the prevalence of inflammatory bowel disease increased from 25,345 in 2009 to 47,444 in 2016 [4]. Having an ostomy is classified as a disability, and the numbers of ostomates have increased from 14,000 in 2014 to 15,027 in 2018 [5].

Despite the necessity of surgery for future survival and the advances made in this type of surgery, upon receiving an ostomy, patients must undergo the complex process of adapting to a second period of toilet training in adulthood [6]. Ostomy formation surgeries often introduce various ostomy-related stressors, including body image disturbances and changes in the anatomy and function of the gastrointestinal tract, leading to maladaptation, psychological disturbance, and impaired daily functioning [7,8,9,10]. The changes accompanying an ostomy have a heavy impact on the affected individuals’ quality of life throughout the illness experience [11,12,13].

Coping is defined as the constant effort to manage various internal and external demands (i.e., stressful events) [14]. An individual’s experience of stress largely depends on their appraisal of a situation that exceeds their capability to cope [14]. Coping is the primary positive task for patients in illness experiences and is related to physical, psychological, social, sexual, and spiritual problems [6,15,16]. The primary areas of difficulty for people who are coping with ostomies include: (1) uncontrollable bowel functioning (physical); (2) depression, distress, and anxiety due to disturbed body image (psychological); (3) occupational dysfunction (economic); (4) reduced social relations and activities (social); (5) decreased intimacy and sexual dysfunction (sexual); and (6) religious issues (spiritual) [1,2,11,15,17]. Various coping strategies are used to deal with such problems. For example, social support for patients with ostomies has been shown to be a significant factor in improved coping [16]. In particular, adaptive or positive coping appears to be associated with better physical health, fewer psychosocial problems (e.g., depression and anxiety), less social disruption, less sexual dysfunction, and high spiritual well-being [6,18,19,20].

When supporting patients with ostomies in coping with the effects of their disease, healthcare professionals must remember that they are dealing with an illness. A disease refers to a specific pathology within the body, whereas an illness concerns patients’ subjective experiences of symptoms and their psychosocial influences [21,22]. This distinction implies that individuals diagnosed with the same disease may experience different illness trajectories. Therefore, healthcare professionals should explore individual patients’ processes for coping with their illness—from the beginning of its diagnosis to treatment and survivorship [11,23]. Unfortunately, few standardized care plans exist to support coping or adaptation to ostomy throughout the illness process. Obtaining an in-depth understanding of the experiences and suffering of patients with ostomies is necessary to begin developing such care plans.

The qualitative research on the experiences of patients with ostomies [9,11,16,24,25], generally, has not focused on how patients and their family members coped with their experiences of getting an ostomy (e.g., the associated bodily changes), from the time of the patient’s diagnosis and recovery to their daily life after the ostomy formation surgery. Therefore, the purpose of this study is to explore, through a qualitative approach via focus-group interviews, the meaning of the illness—and the related coping experiences throughout the illness trajectory—among individuals with ostomies. The findings of this study might be beneficial for designing integrated interventions that consider individual support preferences of people with ostomies and their families.

## 2. Materials and Methods 

### 2.1. Design

An exploratory qualitative design involving focus group interviews was used to describe the participants’ meaning of illness and interpret their coping experiences. 

### 2.2. Participants

Purposive sampling was utilized to recruit participants through an ostomy association in Seoul, South Korea. Potential participants included all members affiliated with the association who had undergone a colostomy or ileostomy. At the end of the association’s monthly meeting, the principal investigator explained the purpose and detailed procedures of the study to the members attending the meeting. The researcher asked those who were interested and willing to participate in the study to provide their written informed consent to engage in a focus group discussion exploring their lived experiences and to provide basic information, including their phone number, name, gender, age, type of ostomy, and the year of surgery. The research assistant called the participants to schedule the focus group interviews. The study included 19 participants.

### 2.3. Data Collection

Data were collected through focus group interviews from March through May 2017. The participants were split into three groups to ensure adequate diversity. Focus group interviews were conducted to improve the understanding of how those with ostomies cope over time after stoma formation. Focus group interviews have been acknowledged to be effective in providing information on a range of ideas and feelings, thereby illuminating differences in perspective between groups of individuals [26]; they are also effective in generating a group interaction [27].

The research team then conducted two focus group interviews with the group until data saturation was achieved. The interview questions were developed following the questioning route suggested by Krueger and Casey [28], consisting of five steps: opening, introductory, transition, key, and ending questions. Key questions were developed using Corbin and Strauss’s Chronic Illness Trajectory Framework [29], which lists eight status changes that individuals can experience during a chronic condition. Since the study participants were already mid-way through the illness process, six phases were included in the questions and analysis. The six phases were then categorized into three phases to reflect participants’ experiences, starting from the initial diagnosis (Table 1). Further detailed and specific questions were included in the second focus group interview as the first interview’s discussion was insufficient to reach response saturation.

The focus groups were moderated by the principal investigator (HS) with assistance from the coauthor (YK), and the same guidelines and questions were used in both focus group interviews. The moderator’s role was to balance participation and address group dynamics by encouraging all participants to join in the discussion [30,31]. Each interview lasted no more than two hours. The interviews were held in a conference room at the ostomy association. The same research assistants observed the interviews and wrote down participants’ non-verbal expressions throughout data collection, and all interviews were audiotaped with participants’ consent.

### 2.4. Data Analysis

The transcribed interviews were analyzed and presented as words and themes to interpret the results using qualitative content analysis [32]. Following the procedure described by Graneheim and Lundman [33], the recorded interview data were transcribed verbatim by one author (HS). Then, both authors (HS and YK) independently read each transcript repeatedly to obtain a sense of the discussion as a whole, while each author separately classified the statements. After the text about the participants’ experiences of living with an ostomy was extracted and compiled, meaning units, codes, and subcategories were created. The authors repeatedly reviewed and discussed their judgments until they reached an agreement on how to sort the codes. Finally, three themes and nine categories emerged.

### 2.5. Trustworthiness

To increase credibility and achieve trustworthiness, we sought to maximize participant diversity [33]. The 19 participants were divided into three groups that included six to seven participants to ensure adequate diversity in terms of age, gender, cancer diagnosis, ostomy type (i.e., colostomy vs. ileostomy), and time since stoma formation. Additionally, we sought agreement among coresearchers, experts, and participants to present representative quotations from the transcribed texts [33]. During data collection, two research assistants scheduled the focus group meetings and communicated with the participants; during the interviews, they recorded all the statements and nonverbal expressions (e.g., emotional reactions) of the participants and researchers, which were reflected in the transcription. An expert in qualitative data analysis with more than 10 years of experience in conducting qualitative research independently analyzed the transcriptions at the same time as the two authors. The results were compared to ensure sufficient trustworthiness in terms of credibility and transferability.

### 2.6. Ethical Considerations

The institutional review board of the principal investigator’s institution approved the study (1041078-201701-HRSB-008-01). After the study’s purpose was explained, all participants were informed that they could withdraw at any time. Written informed consent was received from all participants. Participants were asked to wear a number tag during all interviews to protect their privacy, and the corresponding numbers were used during the transcriptions to analyze their responses.

## 3. Results

The characteristics of the interview participants are presented in Table 2. Study participants ranged in age from 57 to 82 years (M = 70.8 years, SD = 7.1). Twelve (63.2%) were men, 17 (89.5%) had a colostomy, and 2 had an ileostomy. The mean time since ostomy formation was 14.3 years (SD = 8.3; range: 1–30). Sixteen (84.2%) participants underwent ostomy formation surgery following a diagnosis of colorectal cancer, and three (15.8%) had an ostomy for other reasons (e.g., bowel inflammatory diseases and genetic problems). The coping experiences of patients with ostomies can be summarized into three interrelated themes: struggling and suffering, learning how to live with the ostomy, and living with the ostomy.

### 3.1. Struggling and Suffering

Early in the ostomy formation process (from diagnosis to surgery), participants suffered from physical pain related to the disease and ostomy management. Many underwent cancer treatment (e.g., chemo/radiation therapies) and suffered side effects (e.g., hernia or adhesion). Being unaccustomed to ostomy management, they struggled with unclean ostomies, pain from the ostomy appliance, and uncontrolled defecation.

“With an ileostomy, when you eat something, it just comes right out...Initially, I got two bags a day, but later, the doctor prescribed four bags a week, which were still not enough for me because I used to excrete three to four times a day. So, I went to the physician and told him straight that he’d better kill me or do something about this” (Participant 6).

Participants experienced a loss of control over normal defecation, which burdened their daily life and ultimately changed their self-image. Participants’ family members, particularly spouses, experienced difficulties coping with the stoma.

“After that... it was so toe-curling. Like, I’m doing something I could never imagine doing. It was so absurd. And I wore that sack for six months and felt like… gosh, perhaps it’s better for me to hang myself on a mountain. I had these sorts of thoughts so many times.” (Participant 2).

“My husband came to me and said…’Don’t you think it’s better to die than to live like this?’ It kept leaking. I still can’t forget about it. Even when he is dying (clamping her lips), I can never forget that… From that point, there was no affection… (In an angry tone) It kept leaking…and he kept saying, ‘why are you living like that?’” (Participant 4)

Participants received negative public attention at the hospital due to the malodorous ostomy, which led to social distress. After ostomy formation, participants had to engage in structural coping within their workplace and society, which led to social isolation.

“When I was in the hospital…that was the beginning stage… Even though the curtains were closed, the smell was so bad…people in other beds escaped, they all left… When the ostomy nurse entered, everyone immediately left…the first few times, people were not aware, so they didn’t talk about it, but once it was opened…it must have stunk. They grabbed their noses, and I couldn’t forget their faces. [I still] can’t forget about it. That’s when I realized this problem is much more difficult than what others have.” (Participant 7).

### 3.2. Learning How to Live with the Ostomy

Compared to the beginning, individuals gradually learned to adapt better as they engaged in self-care. Eventually, they learned the necessary ostomy management skills, how to prepare for ostomy accidents, how to build a life around their ostomy, and how to introduce the necessary changes in their eating habits and sexual activity.

“I always keep black leggings or short flesh-toned [stockings] in my bag (with emphasis). I wear them because only these keep diarrhea in. They don’t let it flow down. That’s why I like them and bring two sets in my bag all the time. Our lives are all oriented toward this stoma. Even if I complete a big task…I think I’ll still spend my life just managing this before dying... “ (Participant 5).

The participants attained a degree of psychological comfort through learning effective ostomy management skills and seeking religious help. Many actively endeavored to cope psychologically, including “pocketing their pride” and striving to be more positive.

A person like me, hot-tempered and straight-forward, is now trying to understand more, trying to yield, trying to lose, because that would make me more comfortable. Otherwise, if I just continue with my old heart in this body, I won’t be able to take it... “(Participant 1).

Participants’ spouses also adapted better and began providing more psychological support. Participants received help and mindful comfort from spouses when psychologically distressed. Spouses also provided healthy meals and cleaned dirty clothes after accidents, which helped the participants feel supported and able to overcome psychological distress.

“My husband said, ‘How hard it must have been to live with such a body…Admirable. You are really admirable’. I can live with that comfort.” (Participant 5).

All participants received both informational and emotional support from peers and the ostomy association. They often developed friendships with other patients, who felt like family and gained confidence and courage. However, they continued to receive negative attention from society due to bias, prejudice, discrimination, and ignorance about “poop bags”. Following ostomy formation, participants experienced limitations in participating in social activities such as going out, traveling, using public baths, working, and dating, which caused them to experience various negative emotions (e.g., feeling insulted, despair, and self-consciousness).

“When we let out gas or feces, we also let out odors. I’ve been insulted many times…I went in and let some gas out but did it slowly (speaking softly) because I didn’t want to bother others. Even so, a young guy at the next cubicle shouted, ‘You, lunatic, what the hell did you eat?’ I was going mad.” (Participant 4).

### 3.3. Living with the Ostomy

As a result of the adaptation process, the participants learned to cope with ostomy management physically. They encountered fewer problems with ostomy care and felt more confident in going out or performing their daily routines (e.g., irrigation). At this point, participants rarely experienced accidents and were able to manage their ostomy calmly.

“So after that, I thought I wasn’t even fated to die. I just lived with having an enema diligently. And after a while, I somehow got a little used to it.” (Participant 3).

Nevertheless, participants still struggled to cope with their changed body image; they lacked self-confidence, experienced psychological withdrawal, and often had a negative body image. Many participants also felt anxious about the recurrence of their disease and death.

“I have to wear this [ostomy bag] and walk around, and nobody would say ‘I’m so proud of wearing this.’ If I tell people that I am wearing this, everyone turns around and walks away. They totally leave me out of their conversation. They probably babble this and that… No, actually, they won’t. They won’t even want to talk to me.” (Participant 3).

In particular, the participants with stoma from cancer felt anxious about their disease recurring and the possibility of death.

Participants experienced ambivalence about their ostomy and their “socially useless” self. They found it difficult to speak to others about their ostomy. They had a form of hidden disability—a “normal appearance” coupled with a shadowy and dark mind. While they felt good about how they resembled ordinary individuals externally, and nobody knew of their ostomy until it was mentioned, participants felt that nobody understood how they suffered.

“It’s the most pitiful and inconvenient thing…but we can’t tell others [about it] … unless it’s one of us… [Well, we just] can’t talk about this. Strangely, it kept stinking. So strange. I went to the bathroom and found nothing wrong. I kept wondering why and then looked here, only to see that there were feces smeared here. So, I tried to wash it all off. But even after washing and changing, it was still smeared… (Angry voice) Where can we talk about this? Ostomy patients like us can’t (continues with emphasis)…" (Participant 5).

Participants expressed concern about the limited social resources in later life, assuming that few nursing homes would opt to care for people with ostomies, like themselves.

“(Sullenly) It’s just…when I am feeling blue, [I think] ‘what is going to happen if I become old and can’t do anything with my own hands?’ (In a trembling voice) That hurts me the most. If I can’ use my hands, I have no one to take care of me…that hurts my heart…” (Participant 4).

“I am worried about us…We live together, but we don’t know our future. Nobody [does]… But with family…we live together, but still may end up in nursing homes. Later. If so…I have heard that they don’t take in patients like us…the nursing homes, I mean. Ostomy patients…I have heard that nursing homes don’t take in ostomy patients. If so, where should we go? (Laughs)” (Participant 5).

## 4. Discussion

We qualitatively explored the coping experiences of individuals with ostomies in South Korea through various stages from the initial diagnosis to long-term survival based on the chronic illness trajectory framework [29]. The first theme, belonging to the trajectory, crisis, and acute phases in the post-diagnosis and surgical periods, was struggling and suffering, which dealt with learning to manage the ostomy. Besides physical distress, participants experienced considerable psychological distress due to an altered body image, consistent with earlier studies [7,11,17,20,34].

The second theme was learning how to live with the ostomy, belonging to the stable phase described by Corbin and Strauss [29]. This theme expressed how participants learned ostomy self-care, including preparation for unexpected expulsion of feces out of ostomies, building their lives around the ostomy, and changing their sexual activities and eating habits (adopting a low-residue diet, eating foods that reduce gas and odor formation, and timing of food consumption for ostomy adaptation [35,36]. Asiedu and colleagues [11] noted that these skills are essential for returning to a normal life or full adult personhood. Participants also strived to better adapt to the ostomy, which brought a sense of comfort. Learning practical skills for ostomy care is known to alleviate psychological distress, uncertainty, insecurity, and anxiety [20,37]. 

Family support is critical for patients to accept their bodies and alleviate their psychological distress, which aligns with the results of past studies [11,21]. Interestingly, we found gender differences in spousal support. Specifically, wives took more responsibility for taking care of patients in terms of their diet, appliance, and stoma than husbands; this fact has not been reported previously. The influence of collectivism and Confucianism, which emphasize family ties and the role of the household in South Korea, might explain this finding [38]. If patients’ families could not provide adequate care for male patients, they might be blamed by society. Further studies should explore other social and cultural factors affecting coping in this population.

The third theme was living with the ostomy, corresponding with the unstable and downward phases of Corbin and Strauss’s [26] conceptual framework. Participants reported that they eventually achieved physical coping through mastery over ostomy management, consistent with most past findings [21,24,36]. However, in contrast to our study, Sun et al. [9] and Grant et al. [35] found that patients with more than five years of ostomy experience continued to have ostomy-related problems (e.g., clothing restrictions, dietary changes, operation of ostomy appliances, self-care, and finding alternative ways to cope with living with an ostomy). This difference might be attributed to the characteristics of the samples in different studies. Sun et al. [9] and Grant et al. [35] recruited samples of long-term colorectal cancer survivors at least 5 years, post-diagnosis, from a variety of regions in the country, while we used a relatively homogenous sample from an ostomy association in Seoul that offers various types of self-care management for their members. Additionally, 15.8% of the interview participants were diagnosed with a non-cancer disease (i.e., inflammatory bowel disease). These sample differences might contribute to variations in findings regarding physical coping with an ostomy.

In contrast to positive physical coping, participants reported a degree of psychological maladaptation. They described struggles with their body image, which led to low self-confidence, psychological atrophy, and fixation on a negative body image. Additionally, participants reported a fear of recurrence and death. Previous studies have shown that cancer survivors tend to have a higher prevalence of depression than those who have never had cancer [34,39]. However, in this study, we did not compare differences in depression or anxiety between patients with and without cancer. Therefore, further research is needed to compare the prevalence of depression and anxiety among cancer survivors to that among patients with inflammatory bowel diseases.

Negative body image was a persistent problem, although it showed some improvement in coping. Other studies have shown that even after physical-symptom-related distress improved among patients with ostomies, many remained psychologically distressed by a disturbed body image [7,34]. Factors related to this disturbed body image have, unfortunately, yet to be clarified. Further research should explore these factors to identify individuals at risk for maladaptive body changes. Additionally, modifiable factors should be corrected to ensure better coping at the individual and family levels. At the societal level, it might help to disseminate more health information about ostomy to familiarize the public with ostomies, as doing so may facilitate individuals’ coping ability.

Participants reported fear of harm to their social reputation and loss of dignity due to a malodorous ostomy. Other studies have reported similar findings [25,35,40]. Two cultural concepts may explain patients’ social coping with ostomies in both Western society and South Korea: the hierarchy of dirt and social competence in full adult personhood. As for social competence in full adult personhood, bladder and bowel control are considered the most basic physiological abilities. Accidents, noises, and odors related to ostomies make an inherently private issue public, leading to participants being viewed as socially incompetent [24,36], as they have lost control over bowel movement. Culturally, defecation in improper places and times is unacceptable because of a nexus between dirt and impairment [41,42]. Cultural differences regarding disabilities could also explain participants’ social maladaptive coping. In South Korea, having a disability is regarded as abnormal, and those with disabilities are stigmatized and marginalized by society [43]. Although an ostomy is invisible to others, it is frowned upon by society.

Participants also discussed the challenge of having limited social resources, which has not previously been reported. In South Korea, there are no locations or toilets for patients with a stoma in subways or public places. In contrast, in other countries such as England and Japan, public toilets are available for changing stoma devices. Further, older adult patients with an ostomy have described difficulties finding a living place, such as a long-term care facility, when they become dependent. Ironically, older adults living with a stoma represent a problem, owing to the lack of social resources such as nursing homes or institutions to tend to stoma patients. These cultural differences in the perceptions of people with ostomies are pivotal in their psychological coping in South Korea.

Unlike past studies, in this study, we explored the coping process of individuals with ostomies right from their ostomy surgery, based on a conceptual framework. Despite this strength, the study has some limitations. First, the findings should be interpreted with caution when applying the results to other populations, because the participants in the sample showed notable variation in the duration for which they had lived with an ostomy, and because the authors could only conduct two focus groups interviews. The fact that the participants were recruited from an ostomy association might have contributed to their stoma formations over a wide range of time because they had difficulty engaging in social activities during the acute stage. In addition, there may be more individuals who had stoma surgery a long time ago since the ostomy association also functions as an opportunity to meet people and develop friendships, which may indicate the participants share similar characteristics in terms of a highly selected sample of survivors in the same association. Therefore, it is necessary to recruit the participants from various associations in the future. Additionally, further research should focus on recruiting participants at a similar illness stage (i.e., time with an ostomy) to validate the study’s findings. Second, we did not compare differences in coping experiences between participants with cancer and those with inflammatory bowel diseases. Compared to those with a stoma from non-cancer disease (i.e., inflammatory bowel disease), those with a stoma due to cancer may experience more difficulties during the active treatment period since they must cope with both the ostomy and cancer treatment. However, previous research found that patients with cancer did not have difficulty coping with stomas because they were viewed stoma as salvation and chance for survival [44]. Accordingly, future research should examine the differences in coping between the two groups. Finally, we used the Corbin and Strauss’s Chronic Illness Trajectory Framework with eight status changes during a chronic condition, but we included six phases in the questions and analysis excluding two phases of initial and dying phases since the current study focused on post surgery survivors. Therefore, the study results limit the full application of the framework. Further research is needed to reflect the application of eight stages of the framework.

This study has some implications for practice, education, and policy. In healthcare practice, healthcare professionals can help patients coping with an ostomy by considering their spousal support and sociocultural context when designing interventions. These findings might also help develop systematic programs for patients and their spouses. In terms of education, understanding patients’ coping stages could assist with the development of educational materials for this population. Furthermore, the results have policy implications, and support strategies for people with ostomies should be implemented at the national level. For example, social campaigns for raising public awareness of the challenges faced by those with an ostomy may help patients’ in their social coping.

## 5. Conclusions

We found that, in contrast with physical coping, psychological and social coping remained constant problems throughout the coping process for both patients and their spouses. The conceptual framework used in this study was beneficial for understanding the individual, family, and social support experienced by patients with an ostomy. Interventions targeting family members should focus on the facilitation of coping with ostomy while maintaining the integrity of the family dynamic for both patients’ and their families’ psychological health. The themes identified suggest that gender should be considered when designing interventions to facilitate patients’ coping with ostomies. 

## Figures and Tables

**Table 1 healthcare-09-00021-t001:** Focus group interview questions.

Phases [29]	Key Questions
Trajectory Onset, Crisis, and Acute Phases in the post-diagnosis and surgical periods	Please tell me about your experiences and feelings after the surgery of stoma formation. What were the responses of your family after the surgery of stoma formation? What were your feelings from the response? Please tell me about your experiences and feelings in the hospital.
Stable Phase	What did you do to adapt to your stoma? What were the responses of your family to the adaptation of your stoma formation? Please tell me about your experiences during the adaptation period of stoma.
Unstable and Downward Phases	Please tell me how you have undergone and coped with your life with your stoma. How would you describe your life before and after the surgery? How did you feel? Please tell me any difficulties in physical/mental/social status characterized by your increased concern.

**Table 2 healthcare-09-00021-t002:** Characteristics of informants (*N* = 19).

Characteristics	Frequency (%)	Mean (SD)	Range
Age		70.8 (7.1)	57–82
Gender			
Male	12 (63.2)		
Female	7 (36.8)		
Educational level			
Middle school	9 (47.4)		
High school	8 (42.1)		
College or above	2 (10.5)		
Diagnosis			
Colorectal cancer	16 (84.2)		
Inflammatory disease	3 (15.8)		
Stoma types			
Colostomy	17 (89.5)		
Ileostomy	2 (10.5)		
Years since stoma formation		14.3 (8.3)	1–30

## Data Availability

The data that support the findings of this study are available on request from the corresponding author, [YK]. The data are not publicly available due to restrictions (e.g., they contain information that could compromise the privacy of research participants).

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
