# Peer review of "Coping Processes of Patients with Ostomies in South Korea: A Focus Group Study"

_healthcare, 2020, doi:10.3390/healthcare9010021_

Round 1

Reviewer 1 Report

The authors have addressed the points as suggested in the earlier version, the overall presentation of this manuscript is good. I have no further comment.

Author Response

Response to Reviewer 1 Comments

Point 1: The revised manuscript properly addresses all the comments previously suggested. In my opinion, this improved version is in conditions to be published with no further need for review.

Response 1: We appreciate your positive feedback on the revised paper.

Reviewer 2 Report

The revised manuscript properly addresses all the comments previously suggested. In my opinion, this improved version is in conditions to be published with no further need for review.

Author Response

(The authors gave the same response as above.)

Reviewer 3 Report

The authors attempted to address each comment, and I think the paper is improved in a number of ways. For example, the addition of stoma statistics to situate the study and the table with participant information was helpful. I also appreciated efforts to clarify the sampling limitations (including the note that this is an exploratory study) and the use of the trajectory model.

The revision addressed the Corbin and Strauss framework that was used, but the revision is a bit incoherent.  At 2.1 Design the first sentence is revised but the rest of the paragraph is the same as before.  An exploratory qualitative design would be a grounded theory approach, and yet the next sentence is re-imposing the idea of a framework.  It should be one or the other.

Another area that is not coherent is 2.3 Data collection, where the framework of 8 statuses (only 6 are used), are said to be condensed to 3 patterns, but Table 1 calls them patterns, not phases, quite confusing.

The three Results sections do not seem to relate to the 3 patterns in Table 1.

Overall, three chief concerns remain: 1) whether there should be a framework or a grounded theory approach to the data collected, 2) the appropriateness of the method of collection, focus groups from a highly selected sample of survivors and association members, and 3) key differences within the sample that are not meaningfully analyzed and discussed. How do patients’ age, time since stoma formation, health status and underlying disease diagnosis, membership in a stoma association, etc. impact their experiences? These variables seem integral to the illness trajectory model being used. Without them, I do not see how this framework relates to the data and think the authors should reassess their analytical approach.

Given that this is a fundamental problem with the way the study appears to be conceptualized, I do not know that the paper is suitable for publication.

Author Response

Dear Reviewer:

Thank you for the opportunity to revise and resubmit the manuscript titled “The Process of Coping for Patients with Ostomies in South Korea: A Focus Group Study” to the Healthcare.

We very much appreciate the thoughtful reviews our manuscript reviewed and feel that the manuscript has further improved by their suggestions. We were happy that each reviewer found the paper greatly improved. We have responded to the suggestions as follows: 

We have marked the revisions made to the manuscript red font. We hope the revised manuscript addresses all of their concerns and comments.
We look forward to hearing from you. Thanks.

Respectfully,

This manuscript is a resubmission of an earlier submission. The following is a list of the peer review reports and author responses from that submission.

Round 1

Reviewer 1 Report

Review of “Coping Processes of Patients with Ostomies in South Korea: A Focus Group Study”

Healthcare ID 2020, 8, x; doi: FOR PEER REVIEW

This paper reports on analysis of data from two focus groups of patients in South Korea who were coping with ostomies following colon cancer or bowel disease diagnoses and treatment.  The researchers were interested in the ways in which patients and their families coped with ostomies from the initial surgery through the period to adaptation to daily life.  The sample was made up of 19 ostomy patients who were split into three groups, each interviewed two times.  The researchers used the Corbin and Strauss Chronic Illness Trajectory Framework with exploratory qualitative methods.  They identified three related themes of: struggling and suffering, learning how to live with the ostomy, and living with an ostomy. This is an important topic that, to my knowledge, has yet to receive sufficient coverage in the literature (and in particular, this journal). Their findings also hold implications for policy, such as the need for more inclusive restrooms that accommodate people needing to change a stoma.

  1. The paper begins by mentioning the worldwide increase in ostomy creation surgeries. It would be helpful to cite some specific prevalence statistics and the trends, especially for South Korea, to provide context to justify the research. Have there been advances in this type of surgery?

  1. The sample was recruited from an ostomy association in South Korea. What selection factors might have biased the sample? For example, are there statistics on the age range, or the gender distribution of all patients receiving such surgery that could be compared to the sample?  Is there a reason that family members were not interviewed?

  1. The text and table at the top of p. 3 were confusing. The researchers claim to be using the Corbin and Strauss Chronic Illness Trajectory framework, but they state that this framework has eight status changes, and they only include six, but they categorize them into three “trajectories”. It seems incorrect to call the three phases listed in Table 1 “trajectories”, because that implies an arc or pattern that takes place over time.  The end of life trajectories described by Lunney et al. are a good example: sudden death, terminal illness, chronic illness, frailty.  Each of these covers the period from diagnosis to death.  In Table 1 it appears that the three trajectories are just segments of the post-surgical period.  The term Phases is used – that would be a more appropriate term for stages that follow one another.  It might also be helpful to link these phases to the length of time since surgery.

  1. At first glance, using the trajectory model seemed fitting for this study because it would demarcate differences in experience among patients depending on the phase of their illness and the timing of their stoma surgery. However, key variables such as the participants’ ages, time since stoma formation (1 to 30 years is a huge range!), the underlying disease that led to their need for an ostomy in the first place (e.g., cancer vs. inflammatory bowel disease or some other diagnosis), as well as their illness progression (e.g., stage of cancer) and overall health status (stoma notwithstanding) do not seem to be meaningfully integrated into the analysis. But these factors seem critical in that they heavily influence both the experiences and needs of stoma patients! Moreover, excluding them seems to go against the primary rationale for using an illness trajectory model in the first place. For example, reading the observation on page 6, line 226 that “many participants also felt anxious about the recurrence of their disease and death” followed by a participant quote about cancer patients on struggling to think positively, I am left wondering whether this has anything to do with stoma coping (versus an experience of those living with cancer), and relatedly, whether this sentiment would apply at all to those with stomas who do not have cancer. I think conceptual and analytical clarity is needed to parse out the meanings in these observations.

  1. A related but distinct point is that the “three themes” that the authors claim to find in their analysis do not seem particularly tethered to the data describe. I think again, that some of this goes back to the adoption of the trajectory model (or rather, the attempt to impose illness phases into the analysis without critical variables). Instead, what I’m finding is that the framework distracts from what could be standalone themes, such as recurrent talk of suicide, feelings of shame about odor and body image, both the presence of social support as well as the absence of people to talk to who could understand their experiences, etc. These themes—in contrast to the ones given—seem more coherent to me, especially considering the excerpts and explanations provided. Perhaps these were the “nine categories” that the authors mentioned (page 3, line 125) but then did not enumerate?

  1. Support for and organization of arguments: Sometimes themes appeared in the interview quotes but were not actively discussed in the paper while other discussion points were not supported by excerpts but perhaps should have been. There were also times when the discussion did not seem to match the participant quote. For example, the quote “perhaps it’s better for me to hang myself on a mountain” (page 5, line 170) was immediately preceded by a line about family members experiencing difficulties coping with the stoma. The quote did not seem to relate to family challenges at all, nor did the discussion note the suicidal ideation present in the quote. This disrupts the logical flow of the paper. Since there are no restrictions on length of manuscripts for this journal, it seems like the authors could expand their findings and discussion to include more examples (i.e., interview excerpts) and thorough explanations of their analysis.

  1. Although both gender and age differences were mentioned in the abstract, only gender differences in support and caregiving were given much attention in the results.

  1. By definition, the participants in the study were long-time survivors, with on average 14 years since their surgery. How unusual are these survival rates (another issue of selection that should be considered).

Author Response

Response to Reviewer 1 Comments

Point 1: This paper reports on analysis of data from two focus groups of patients in South Korea who were coping with ostomies following colon cancer or bowel disease diagnoses and treatment.

The researchers were interested in the ways in which patients and their families coped with ostomies from the initial surgery through the period to adaptation to daily life. The sample was made up of 19 ostomy patients who were split into three groups, each interviewed two times.

The researchers used the Corbin and Strauss Chronic Illness Trajectory Framework with exploratory qualitative methods. They identified three related themes of: struggling and suffering, learning how to live with the ostomy, and living with an ostomy.

This is an important topic that, to my knowledge, has yet to receive sufficient coverage in the literature (and in particular, this journal). Their findings also hold implications for policy, such as the need for more inclusive restrooms that accommodate people needing to change a stoma.

The paper begins by mentioning the worldwide increase in ostomy creation surgeries. It would be helpful to cite some specific prevalence statistics and the trends, especially for South Korea, to provide context to justify the research. Have there been advances in this type of surgery?

Response 1: Thanks for your encouraging feedback and appreciating the originality and importance of this study.

In response to your feedback about the global prevalence, we have added the information about the prevalence of colorectal cancer, inflammatory bowel disease, and the numbers of individuals living with ostomies in South Korea in the Introduction (page 1, lines 32–36).

Point 2: The sample was recruited from an ostomy association in South Korea. What selection factors might have biased the sample? For example, are there statistics on the age range, or the gender distribution of all patients receiving such surgery that could be compared to the sample? Is there a reason that family members were not interviewed?

Response 2: Thank you for raising this valuable point. We agree with you that the fact that the sample was recruited from the ostomy association might have biased the sample. Specifically, using the association for recruitment may have led to the wide range of time since the participants’ stoma formation. We have revised the Discussion section to include this potential bias (page 8, lines 345–349).

Point 3: The text and table at the top of p. 3 were confusing. The researchers claim to be using the Corbin and Strauss Chronic Illness Trajectory framework, but they state that this framework has eight status changes, and they only include six, but they categorize them into three “trajectories”. It seems incorrect to call the three phases listed in Table 1 “trajectories”, because that implies an arc or pattern that takes place over time. The end of life trajectories described by Lunney et al. are a good example: sudden death, terminal illness, chronic illness, frailty. Each of these covers the period from diagnosis to death. In Table 1 it appears that the three trajectories are just segments of the post-surgical period. The term Phases is used – that would be a more appropriate term for stages that follow one another. It might also be helpful to link these phases to the length of time since surgery.

Response 3: Thank you for this feedback, and we apologize for any confusion. The framework that we used in this study did not cover all of the trajectories that were identified in Corbin and Strauss’s framework; however, we agree with you that we could have presented the different types of categorization more clearly. To address this issue, we have revised a sentence in the text and Table 1 (page 3, lines 104–105).

Point 4: At first glance, using the trajectory model seemed fitting for this study because it would demarcate differences in experience among patients depending on the phase of their illness and the timing of their stoma surgery. However, key variables such as the participants’ ages, time since stoma formation (1 to 30 years is a huge range!), the underlying disease that led to their need for an ostomy in the first place (e.g., cancer vs. inflammatory bowel disease or some other diagnosis), as well as their illness progression (e.g., stage of cancer) and overall health status (stoma notwithstanding) do not seem to be meaningfully integrated into the analysis. But these factors seem critical in that they heavily influence both the experiences and needs of stoma patients! Moreover, excluding them seems to go against the primary rationale for using an illness trajectory model in the first place. For example, reading the observation on page 6, line 226 that “many participants also felt anxious about the recurrence of their disease and death” followed by a participant quote about cancer patients on struggling to think positively, I am left wondering whether this has anything to do with stoma coping (versus an experience of those living with cancer), and relatedly, whether this sentiment would apply at all to those with stomas who do not have cancer. I think conceptual and analytical clarity is needed to parse out the meanings in these observations.

Response 4: Thank you for raising this valuable point. We agree with your comment that the individuals with stomas from non-cancerous diseases and those with stoma from cancer may have differences in coping. We also agree that this should be addressed directly in the manuscript, so we have revised the sentence and included the quote (lines 252-258). We have also included text regarding these group differences in the Discussion section (page 9, lines 352-357).

Point 5: A related but distinct point is that the “three themes” that the authors claim to find in their analysis do not seem particularly tethered to the data describe. I think again, that some of this goes back to the adoption of the trajectory model (or rather, the attempt to impose illness phases into the analysis without critical variables). Instead, what I’m finding is that the framework distracts from what could be standalone themes, such as recurrent talk of suicide, feelings of shame about odor and body image, both the presence of social support as well as the absence of people to talk to who could understand their experiences, etc. These themes—in contrast to the ones given—seem more coherent to me, especially considering the excerpts and explanations provided. Perhaps these were the “nine categories” that the authors mentioned (page 3, line 125) but then did not enumerate?

Response 5: We agree with you that the themes that emerged through our analysis may not fit perfectly with the trajectory model, possibly owing to the recruitment source for the study participants. However, we do think that the model has a good fit with the primary focus of the study regarding the ongoing process of coping and adaptation for individuals with stomas.

Thank you for this feedback regarding the nine categories that we did not enumerate. We apologize for this oversight and have removed reference to the nine categories from the text.

Point 6: Support for and organization of arguments: Sometimes themes appeared in the interview quotes but were not actively discussed in the paper while other discussion points were not supported by excerpts but perhaps should have been. There were also times when the discussion did not seem to match the participant quote. For example, the quote “perhaps it’s better for me to hang myself on a mountain” (page 5, line 170) was immediately preceded by a line about family members experiencing difficulties coping with the stoma. The quote did not seem to relate to family challenges at all, nor did the discussion note the suicidal ideation present in the quote. This disrupts the logical flow of the paper. Since there are no restrictions on length of manuscripts for this journal, it seems like the authors could expand their findings and discussion to include more examples (i.e., interview excerpts) and thorough explanations of their analysis.

Response 6: Thank you for pointing this out. We have included a quote to support the difficulty of family member (page 5, lines 174-178).

Point 7: Although both gender and age differences were mentioned in the abstract, only gender differences in support and caregiving were given much attention in the results.

Response 7: Thank you for this feedback. In response, we have included more quotes to provide examples of the issues related to later life and aging identified by participants (page 7, lines 253-263).

Point 8: By definition, the participants in the study were long-time survivors, with on average 14 years since their surgery. How unusual are these survival rates (another issue of selection that should be considered).

Response 8: Our study included participants who were individuals with ostomies due to cancer and non-cancer diseases; thus, we think the survival rates for the participants is less important in this study than one that includes only a sample of cancer survivors. With regard to selection issues, we have identified these aspects of recruitment and sample characteristics in our review of the study limitations (page 9, lines 352–357).

Reviewer 2 Report

The paper focus an interesting topic and is very well written. The context of the manuscript is properly set up, being cited key authors in the introductory text.

Other comments:

Line 43: The authors present a definition of ostomy that should be introduced earlier in the text, namely in the first sentences when the problematic of the manuscript is addressed.

Line 44: The authors say “It is typical for people coping with ostomies to focus on addressing the following problems”. However, it is difficult to understand what do the authors meant to say with this. These are the main areas where people have most difficulties to cope with ostomies? These are the main consequences of ostomies? Please clarify

Line 49: Can the authors give some examples of the “Various coping strategies” “used to deal with such problems ”?

Line 52: Can the authors explain how these cross-sectional studies have “captured the relationships between the above problems among patients with ostomies or identified the factors related to coping with an ostomy “?

Line 55: Why nurses? And what is the role of other health professionals in these process

Line 65 to 73: Authors refer that, contrary to other studies published in the area, their focus is on the patient and his/her family (and ultimately, in the nurses accompanying them throughout the illness process). Attending to this, it seems that not only a person-centered care perspective is explored in the manuscript, but a relationship-centered care point of view.

Please see as examples the following references:

- HAIDET, Paul; STEIN, Howard F. The role of the student-teacher relationship in the formation of physicians. Journal of General Internal Medicine, 2006, 21.1: 16-20.

- DUPUIS, Sherry, et al. Theoretical foundations guiding culture change: The work of the Partnerships in Dementia Care Alliance. Dementia, 2016, 15.1: 85-105.

- SLINGSBY, Brian Taylor. Professional approaches to stroke treatment in Japan: a relationship‐centred model. Journal of Evaluation in Clinical Practice, 2006, 12.2: 218-226.

Line 74: For the section Materials and methods, please follow the SRQR or COREQ guidelines (https://www.equator-network.org/). Overall details are described but there are some minor information missing or needing minor clarification.

Line 271: Can the authors provide some possible explanations for the differences observed in their study and in the studies of Sun et al. [5] and Grant et al. [34]?

Line 323: The authors mention the role of nurses but this is not properly address in the results. Why choosing to mention only this group of health professionals? The same happens for the role of the family which is only discussed in general terms and then mentioned in the conclusions as an important topic.

Moreover, the patient-centered care mentioned in the end of the Introduction is not addressed in the rest of the manuscript. So, what is the relevance of it?

Author Response

Response to Reviewer 2 Comments

Point 1: The paper focus an interesting topic and is very well written. The context of the manuscript is properly set up, being cited key authors in the introductory text.

Other comments:

Line 43: The authors present a definition of ostomy that should be introduced earlier in the text, namely in the first sentences when the problematic of the manuscript is addressed.

Response 1: Thank you so much for your kind words and support. We appreciate the time you took to review our manuscript.

We agree that a definition of ostomy should be introduced earlier in the Introduction. In our revision, we have moved the definition of ostomy to the first sentence of the introduction (page 1, lines 29–30).

Point 2: Line 44: The authors say “It is typical for people coping with ostomies to focus on addressing the following problems”. However, it is difficult to understand what do the authors meant to say with this. These are the main areas where people have most difficulties to cope with ostomies? These are the main consequences of ostomies? Please clarify

Response 2: Thank you for bringing this to our attention—we apologize for any confusion. In response, we have revised the sentence to improve its clarity (page 2, line 48).

Point 3: Line 49: Can the authors give some examples of the “Various coping strategies” “used to deal with such problems ”?

Response 3: Yes, we agree that adding examples of these coping strategies is needed, and we have now provided an example of a coping strategy (page 2, lines 52–54).

Point 4: Line 52: Can the authors explain how these cross-sectional studies have “captured the relationships between the above problems among patients with ostomies or identified the factors related to coping with an ostomy “?

Response 4: Thank you for this comment. Upon reflection, we have decided that this sentence might mislead the audience; therefore, we have deleted it.

Point 5: Line 55: Why nurses? And what is the role of other health professionals in these process.

Response 5: We expanded the wording in the manuscript to include the role of other healthcare professionals rather than limit our focus to nurses throughout the paper.

Point 6: Line 65 to 73: Authors refer that, contrary to other studies published in the area, their focus is on the patient and his/her family (and ultimately, in the nurses accompanying them throughout the illness process). Attending to this, it seems that not only a person-centered care perspective is explored in the manuscript, but a relationship-centered care point of view.

Response 6: Thanks for your thoughtfulness in providing these reference articles.

However, we decided to delete the term patient-centered care because we could not find any themes relevant to patient-centered care.

Point 7: Line 74: For the section Materials and methods, please follow the SRQR or COREQ guidelines (https://www.equator-network.org/). Overall details are described but there are some minor information missing or needing minor clarification.

Response 7: Thank you for pointing this out. Following the SRQR guidelines and checklist, we carefully reviewed the Methods section and included information on the qualitative approach in the design subsection (page 2, lines 77–78).

Point 8: Line 271: Can the authors provide some possible explanations for the differences observed in their study and in the studies of Sun et al. [5] and Grant et al. [34]?

Response 8: Thank you for this recommendation. Yes, we have added to the Discussion to provide an explanation about the differences between these studies and our study (pages 7–8, lines 294–301).

Point 9: Line 323: The authors mention the role of nurses but this is not properly address in the results. Why choosing to mention only this group of health professionals? The same happens for the role of the family which is only discussed in general terms and then mentioned in the conclusions as an important topic.

Response 9: We agree with your point regarding nurses and family. In response, we revised the manuscript by expanded the phrases regarding the roles of nurses to healthcare professionals. Also, we focused the wording to be on spouses rather than all family members, which is more consistent with our findings.

Point 10: Moreover, the patient-centered care mentioned in the end of the Introduction is not addressed in the rest of the manuscript. So, what is the relevance of it?

Response 10: We wish the findings of this study were relevant to patient-centered care, but we could not find evidence of it. Thus, we made a decision to delete references to patient-centered care from the Introduction

Reviewer 3 Report

Son and Kang, conducted a questionnaire-based study to identified possible complications post ostomy and how patient’s coping in this condition. This manuscript is well conceptualized and may help the policy makers to design some strategies for the welfare of patients. Major concern with this study is a low sample size, which authors should discuss in the manuscript. Authors need to address:

  1. A tabular presentation of patients age, sex, type of ostomy and duration with ostomy is required in the manuscript for better understanding.
  2. Data curated from qualitative analysis provided by patients, and there is no doubt the sufferers has a common perspective regarding ostomy, whether authors had any input from healthcare professional as another source of verifying data?
  3. Line 126- 138- it seems authors have inducted multiple assistants to collect and analyze data, are they medical professionals or family members?
  4. Though ostomy imposes a physical and psychological stress to patients but, it is a required surgical process for future survival, authors should explain the necessity of ostomy in the context of post-surgery complications.

Author Response

Response to Reviewer 3 Comments

Point 1: Son and Kang, conducted a questionnaire-based study to identified possible complications post ostomy and how patient’s coping in this condition.

This manuscript is well conceptualized and may help the policy makers to design some strategies for the welfare of patients.

Major concern with this study is a low sample size, which authors should discuss in the manuscript. Authors need to address:

Response 1: Thank you for the positive feedback. We greatly appreciate the time you have taken to review and provide feedback on our manuscript.

We appreciate your feedback regarding the sample size; however, we adapted a qualitative design and used three focus groups to explore the experiences of people with ostomies throughout the illness trajectory. Given that this was a qualitative study, we believe the sample size is sufficient for the study purpose and study design.

Point 2: A tabular presentation of patients age, sex, type of ostomy and duration with ostomy is required in the manuscript for better understanding.

Response 2: We entirely agree with this recommendation, and in response, we created Table 2.

Point 3: Data curated from qualitative analysis provided by patients, and there is no doubt the sufferers has a common perspective regarding ostomy, whether authors had any input from healthcare professional as another source of verifying data?

Response 3: Thank you for this thoughtful question. An expert in qualitative research contributed to the data analysis. The expert was not a healthcare professional, but did have more than 10 years of experience in healthcare field, particularly in the area of nursing.

Point 4: Line 126- 138- it seems authors have inducted multiple assistants to collect and analyze data, are they medical professionals or family members?

Response 4: Thank you for bringing up this important question. We had two research assistants who were enrolled in a master’s in nursing program to assist with data collection. Three individuals, including an expert in qualitative data analysis and the two authors, analyzed the data.

Point 5: Though ostomy imposes a physical and psychological stress to patients but, it is a required surgical process for future survival, authors should explain the necessity of ostomy in the context of post-surgery complications.

Response 5: Thank you for this critical comment. We agree with you and have emphasized the necessity of ostomy in the context of post-surgery complications and survival (page 1, line 37).
